



# Verification of Near Surface Wind Patterns in Germany using Clear Air Radar Echoes

Sebastian Buschow[1] and Petra Friederichs[1]

[1]Institute of Geosciences, University of Bonn, Bonn, Germany

**Correspondence:** Sebastian Buschow (sebastian.buschow@uni-bonn.de)

**Abstract.** The verification of high-resolution meteorological models requires highly resolved validation data and appropriate tools of analysis. While much progress has been made in the case of precipitation, wind fields have received less attention, largely due to a lack of spatial measurements. Clear-sky radar echoes could be an unexpected part of the solution by affording us an indirect look at horizontal wind patterns: Regions of horizontal convergence attract non-meteorological scatterers such as insects; their concentration visualizes the structure of the convergence field. Using a two-dimensional wavelet transform, this study demonstrates how divergences and reflectivities can be quantitatively compared in terms of their spatial scale, (horizontal) anisotropy and direction. A long-term validation of the highly resolved regional reanalysis COSMO-REA2 against the German radar composite RADOLAN shows surprisingly close agreement. Despite theoretically predicted problems with simulations in or near the 'grey-zone' of turbulence, COSMO-REA2 is shown to produce a realistic diurnal cycle of the spatial scales larger than 8km. In agreement with the literature, the orientation of the patterns in both data-sets closely follows the mean wind direction. Conversely, an analysis of the horizontal anisotropy reveals that the model has an unrealistic tendency towards highly linear, roll like patterns early in the day.

## 1 Introduction

Modern numerical weather models at horizontal resolutions on the order of $1 - 10\,km$ are generally believed to be useful, but their added value compared to coarser models is not easy to quantify. On the one hand, the precise placement of very small features continues to be largely unpredictable. In a gridpoint-by-gridpoint comparison, highly resolved models are punished twice for slight location errors in features which coarser models do not attempt to simulate at all. On the other hand, a single error value summarizing the realism of a highly complex meteorological field is not very informative. To address these issues, a large variety of so-called spatial verification techniques has been developed in recent years. A first systematic survey of the field was undertaken in the spatial forecast verification Inter-Comparison Project (Gilleland et al., 2009, ICP). At this point, almost all efforts were focused on the verification of precipitation forecasts, for several reasons: Firstly, the improved representation of convective precipitation was a main incentive for the development of mesoscale weather models. Secondly, the intermittent nature of rain fields makes the aforementioned double-penalty problem particularly obvious. Lastly, radar (and to a lesser degree, satellite) observations readily provide high-resolution spatial observations of precipitation.





The second phase of the ICP project (Dorninger et al., 2018, MesoVICT) has highlighted the need for a spatial verification of other meteorological variables, particularly wind: Wind fields at kilometer resolutions can produce highly complex patterns with potential impacts on convective initiation, wind energy, air quality and aviation safety. The task of verifying spatial wind forecasts poses practical, methodological and theoretical challenges.

From a practical point of view, we face a lack of spatial observations: Model analyses (e.g. used for wind verification by Zschenderlein et al. (2019)) conveniently provide highly resolved, gap-free data but the realism of the underlying model would have to be verified against some other data beforehand. Interpolated station data (for example the VERA analysis used within MesoVICT) are generally too coarsely resolved to represent structures on the scale of single kilometers, denser station networks such as the WegenerNet data-set used by Schlager et al. (2019) are rare. Bousquet et al. (2008) and Beck et al. (2014) use Multi-Doppler wind retrievals from the French national radar network to verify wind predictions from the AROME model. This approach is very appealing but limited to cases with precipitation. In addition, Doppler-derived wind composites are not yet widely available.

Skinner et al. (2016) present a very interesting alternative using single-Doppler azimuthal wind shear as a proxy for low-level rotation. Their study also highlights some of the main methodological challenges related to wind verification: Most spatial verification techniques were developed for scalar quantities which can be decomposed into discrete objects via thresholding.

How should such techniques be adapted to vector fields where non-zero variability is present at every location and the existence of well defined objects is not guaranteed? Skinner et al. (2016), who are interested in tornado forming mesocyclones, chose to focus on the rotational component of the wind field by verifying only the horizontal vorticity. Model and observations are subjected to several spatial filters and then thresholded at manually selected values before the object based MODE technique (Davis et al., 2009) and the image-morphing DAS of Keil and Craig (2009) are applied. Their approach is justified because well-defined objects, i.e., tornadic supercells, clearly exist in the specific case study under consideration. Bousquet et al. (2008) find a similar answer to the vector-problem by verifying horizontal divergences against the corresponding values from the French Multi-Doppler network. Besides point-wise measures, these authors apply a simple scale-separation approach based on a Haar wavelet decomposition of the wind fields. Other recent attempts at spatial wind verification include Zschenderlein et al. (2019) who apply the object-based SAL technique (Wernli et al., 2008) to tresholded predictions of gusts (i.e. absolute wind speed), and Skok and Hladnik (2018) who sort wind vectors into classes based on their speed and direction and use the

popular fractions skill score (Roberts and Lean, 2008, FSS) to find the scales on which the predicted classes agree with the observations.

In this study, we take a similar route as Skinner et al. (2016) but instead of the rotational component we focus on the horizontal divergence of the near surface wind field. Under the right environmental conditions, the spatial pattern of this divergence field can be observed in widely available radar reflectivity data: On warm, rain-free days, convergent boundary layer circulations attract swarms of insects which are drawn in and actively attempt to resist the vertical motion of updraughts (Wilson et al., 1994). The resulting increased concentration of biological scatterers within the radar beam reflects the pattern of convergence and divergence. Numerous studies including Weckwerth et al. (1997, 1999); Thurston et al. (2016); Banghoff et al. (2020) have used this kind of data to study the dominant patterns of boundary layer organization. Atkinson and Wu Zhang





(1996) identified mesoscale shallow convection, organized in the form of cells or horizontal rolls, as the most prominent of those patterns. Numerous studies have used radar data to observe these phenomena (see references in Banghoff et al. (2020)); Banghoff et al. (2020) also present a first long-time climatology using ten years of reflectivities and Doppler velocities from a single radar station in Oklahoma. They manually classified radar images from over 1000 days into cells, rolls and unorganized patterns, reporting organized features on 92 % of summer days without rain. Santellanes et al. (2021) exploited this data-set to

study the environmental conditions that favor the different modes of organization.

  In the present investigation, we aim to study a similarly large data-base of reflectivities from the German RADOLAN-RX composite and compare it to divergence structures from the regional reanalysis COSMO-REA2 (Wahl et al., 2017), covering the time-span from 2007 to 2013. We limit our analysis to small environments around each radar station and consider both the entire COSMO-REA2 time-series (for an overall model climatology) and the subset where clear air radar echoes are available

(for verification).

  For a fair, quantitative validation of the model, the spatial patterns must be analyzed objectively. Here, we rely on the wavelet-based SAD verification methodology of Buschow and Friederichs (2021) which applies a series of directed filters to objectively determine the dominant spatial Scale, Anisotropy and Direction in an image. A closely related approach was used to define a wavelet-based index of convective organization in radar and satellite images by Brune et al. (2021).

To what extent a model at $\mathcal{O}(1km)$ horizontal resolution can be expected to realistically represent boundary layer circulations in the so-called 'Grey-Zone' regime (Wyngaard, 2004) between parametrized and resolved turbulence is a difficult question which poses further theoretical challenges to the verification process. Section 2 therefore briefly summarizes some of the relevant theoretical and experimental results from the literature. Data and methodology are described in sections 3 and 4. Section 5 presents the results of our analysis, including the model-based climatology of divergence structures and its validation

against RADOLAN. Some discussion of our findings is given in section 6, section 7 examines what conclusions can be drawn and identifies avenues for future research.

## 2 Theory and modelling of mesoscale shallow convection

Zhou et al. (2014) have demonstrated how occurrence and basic properties of shallow convective circulation in the atmospheric boundary layer can be understood in analogy to Rayleigh Bénard thermal instability. In the classic framework, the circulation

regime of a fluid between two heated plates is determined by the Rayleigh number

$$\mathrm{Ra} = \frac{g\alpha}{k\nu} \cdot \beta d^4 \,, \tag{1}$$

where $d$ is the distance between the plates, $\beta = dT/dz$ is the temperature gradient, and the coefficients $g, \alpha, k, \nu$ denote gravitational acceleration, thermal expansion coefficient, thermal conductivity and kinematic viscosity, respectively. Eddies of wavelength $\lambda$ start to grow when Ra exceeds a critical value $\mathrm{Ra}_c(\lambda)$. The qualitative sketch in figure 1 shows that this marginal

stability curve has a global minimum near $\lambda = 2d$. For $\mathrm{Ra} < \mathrm{Ra}_c(2d)$, the flow is laminar and heat is exchanged via conduction. When Ra is increased to $\mathrm{Ra}_c(2d)$, convective cells are initiated with a wavelength of roughly twice the depth of the fluid. Zhou

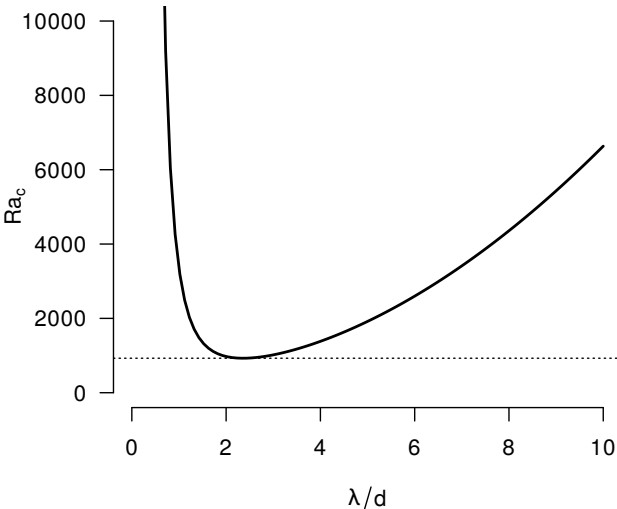

**Figure 1.** Marginal stability curve of Rayleigh-Bénard convection for the classic rigid-rigid boundary conditions. For any given wavelength $\lambda$ (relative to the fluid depth d), Eddies grow if the Rayleigh number lies above the curve and decay otherwise.

et al. (2014) argue that an analogous stability curve applies to the atmospheric boundary layer. In this case, Ra is replaced by a turbulent Rayleigh number of similar form as Eq. 1 wherein the depth $d$ is replaced by the boundary layer height $H$. On a sunny day, the earth's surface is heated and the vertical temperature gradient, as well as the height of the boundary layer increase.

The theory predicts that, once a critical Ra is crossed, the initial wavelength of the circulation should be near $\lambda = 2H \approx 3\,km$; both smaller and larger eddies begin to develop later.

The simulation of this process is challenging because a model with grid-spacing $\delta$ can never resolve eddies with $\lambda < 2\delta$. In large eddy simulations with $\delta << 2H$, convection will correctly be initiated at the natural critical $Ra_c$ with a wavelength of $\sim 2H$. Current NWP models, on the other hand, have $\delta \gtrsim 2H$. In this case, the first eddies to form as Ra increases have

$\lambda \approx \delta$ and initiate at a grid-spacing dependent value $Ra_c(\delta)$. For global or regional models with $\delta \gtrsim 10\,km$, the critical value is so large that such circulations will never form under realistic conditions. Modern mesoscale models, however, operate at $\delta = \mathcal{O}(1km)$ and $Ra_c(\delta)$ becomes attainable. The result is a potentially unrealistic model circulation, the scale and initiation time of which depends on $\delta$. This is one example of the so-called *Terra Incognita* or Grey-Zone of turbulence (Wyngaard, 2004; Honnert et al., 2020), where the highest energy vortices are too large to be adequately represented by the boundary layer

parametrization but too small to be explicitly resolved by the dynamical core of the model. Ching et al. (2014) observed this phenomenon in nested WRF simulations, Poll et al. (2017) detected it in TerrSysMP, the atmospheric component of which is COMSO. Using LES runs of the same models as a reference, both of these studies found that simulations with grid spacing on the kilometer scale initiate turbulence too late and too energetically. In the present study, we will investigate how frequently such small-scaled circulations occur in the climatology of COSMO-REA2 and how they compare to radar observations.





## 3  Data

### 3.1  COSMO-REA2

For a systematic investigation of low-level divergence structures, we ideally need a long, homogeneous time series of high resolution model data. The regional reanalysis COSMO-REA2 is uniquely suited for our need as it provides seven years (2007-2013) of hourly output from the mesoscale model COSMO (Baldauf et al., 2011) at a horizontal resolution of $0.018°$ or roughly 2 km. The model was run with 50 vertical levels over a domain covering Germany and the neighbouring countries. For a full description of the used physics parametrizations, we refer to Wahl et al. (2017) and references therein. For our purposes, it is important to note that boundary layer fluxes are handled by a level-2.5 TKE-closure, shallow convection is parametrized via a modified Tiedtke mass-flux scheme while deep moist convection is left to the dynamic core. The data assimilation uses a continuous nudging scheme to relax the prognostic temperature, wind speed, pressure and relative humidity towards observations from stations, radiosondes, aircraft, ships and buoys. In addition, rain rates from radar observations are assimilated via latent heat nudging (Stephan et al., 2008, LHN). Thus, on clear air days, the only source of mesoscale information (LHN) is inactive, meaning that while data assimilation can help create realistic environmental conditions, the fine-scale structure of the fields is a product of the dynamics and physics of the model. Horizontal divergences were calculated from the hourly 10 m wind vector fields as a simple finite difference approximation.

### 3.2  RADOLAN RX

RADOLAN (Radar online adjustment, 'RADar OnLine ANeichung') RX is the operational radar reflectivity composite of the 16 C-band radars operated by the German weather service. The output has a spatio-temporal resolution of $1\,km \times 1\,km \times 5\,min$ and covers Germany and parts of its neighbours. The underlying radar scans are performed at an orography following elevation angle ($\sim 1°$) with an azimuthal resolution of $1°$ and a range resolution of $250\,m$. Due to the beam geometry, the true native resolution of the reflectivity composite, as well as the height for which it is representative, depends heavily on the distance to the radar station. Pejcic et al. (2020) show that the beams reach typical boundary layer heights of $1 - 1.5\,km$ at about $100\,km$ from the radar location. Therefore, relevant clear-air echoes caused by insects that cannot survive at low temperatures are expected to be found only in the immediate vicinity of the radars.

To get an idea of the type of data we rely on for our model validation, it is instructive to consider an example case. Figure 2 (a) displays the RADOLAN RX composite at noon on 2009-07-29. Aside from a few showers over the North Sea, no appreciable precipitation was observed in Germany on this warm summer day. Temperatures reached values in the high twenties and a high pressure system centred near the German-Polish border generated weak south-easterly flow. Despite the absence of rain, most radars in the composite are surrounded by a disk of low but non-zero reflectivities (-10 to 5 dBZ). While the size, shape and mean intensity of the disks varies, a consistent fine-scaled cellular pattern can be observed throughout central, northern and eastern Germany. Moreover the regions of increased reflectivity are coherently organized in a line-like fashion along a SW/NE direction. Figure 2 (b), showing the corresponding wind and divergence field from COSMO-REA2, reveals that the orientation of the reflectivity lines is broadly consistent with the overall direction of low level flow. Furthermore, the divergence field is

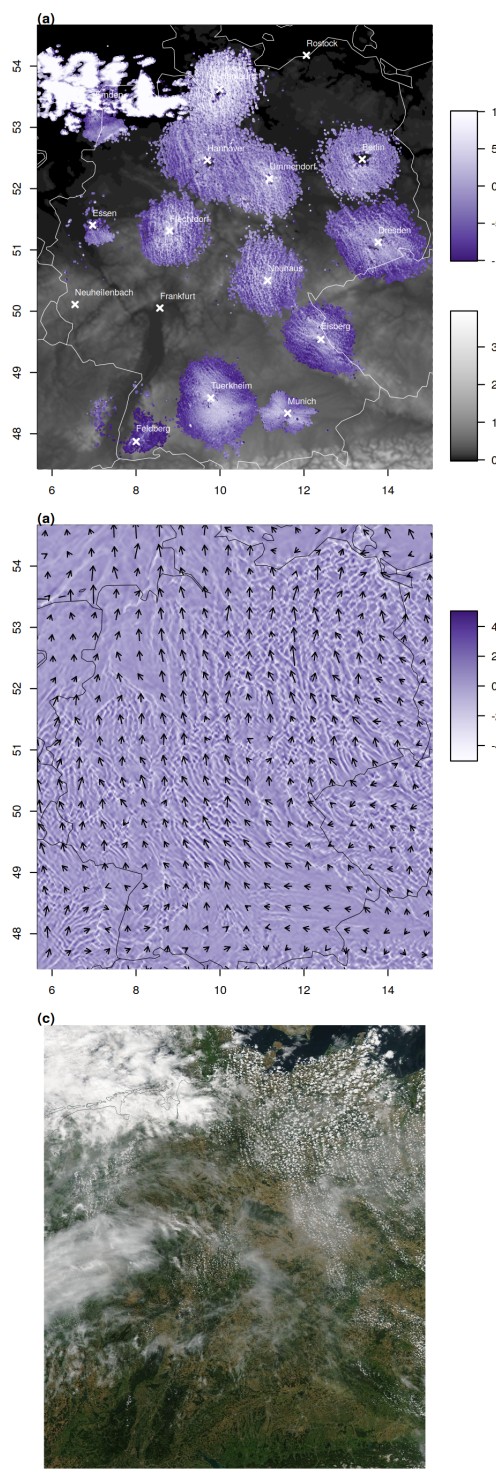

**Figure 2.** RADOLAN RX reflectivity in dBZ (a), COSMO-REA2 10 m divergence (b) and AQUA MODIS satellite image (c) on 2009-07-29 12:00 UTC.



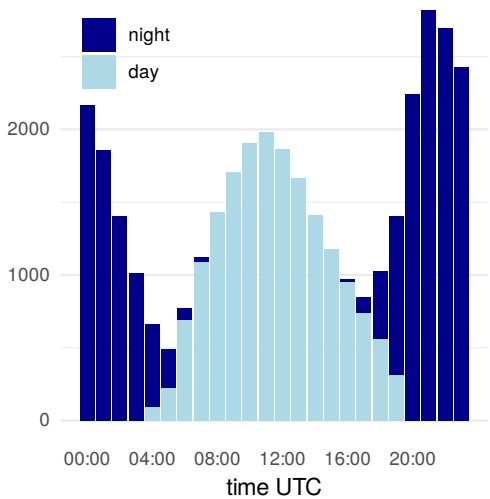

**Figure 3.** Number of complete clear air radar echoes at the twelve selected radars, separately for night and day as defined by sunrise and sunset.

characterized by small scaled patterns of cells and lines with alternating convergence and divergence, the size and orientation of which roughly resembles the radar pattern. Throughout eastern Germany, where the divergences are strongest, the satellite

image in panel (c) shows the typical chains of Cumulus clouds often associated with mesoscale shallow convection (Atkinson and Wu Zhang, 1996). A visual comparison of the reflectivities around, for example, the Berlin radar with the simulated divergences and the clouds in that region leads us to hypothesize that the boundary layer processes in COSMO-REA2 are not entirely unrealistic.

### 3.3 Data availability

As mentioned above, clear-air echoes typically only occur in a small environment around each radar. We therefore limit our study to circular regions with $64\,km$ radius, centred at the 16 radar station which were active throughout the COSMO-REA2 period. While simulated divergences are readily available at every such grid point for each hour between 2007 and 2013, the availability of clear-sky echoes depends on many factors including local topography, technical details of the radars, radar processing at DWD and the life-cycle of the biological scatterers. We consider an individual radar image incomplete if less

than $50\,\%$ of pixels within our $64\,km$ radius around the radar are above $-10\,dBZ$ (visual analysis of many example images has shown that no significant signals exist between roughly $-10\,dBZ$ and the smallest stored value of $-32.5\,dBZ$). From the remaining data, we must filter out rainy episodes, defined here somewhat arbitrarily as cases where at least 100 pixels exceed $+10\,dBZ$. We will refer to all remaining images as *complete*.

Table 1 shows that such complete clear air echoes are overall rare (well below 5 % of all hourly images) and their frequency

varies considerably between radars. For this study, we neglect the four radar stations with the fewest data, thereby removing



**Table 1.** Number of hourly incomplete, rainy, nighttime and complete daytime hourly radar images per station. The top four radars are excluded from further analysis.

|  | incomplete | rain | night | day |
|---|---|---|---|---|
| Frankfurt | 54841 | 6335 | 41 | 104 |
| Emden | 56064 | 5065 | 60 | 132 |
| Essen | 54229 | 6889 | 58 | 145 |
| Rostock | 54627 | 6059 | 295 | 340 |
| Hamburg | 53556 | 6866 | 351 | 548 |
| Munich | 51315 | 9131 | 181 | 694 |
| Feldberg | 52143 | 7419 | 855 | 904 |
| Ummendorf | 52806 | 6847 | 666 | 1002 |
| Neuhaus | 50355 | 8357 | 1527 | 1082 |
| Berlin | 52075 | 6935 | 1011 | 1300 |
| Flechtdorf | 49117 | 9033 | 1830 | 1341 |
| Hannover | 49199 | 8846 | 1478 | 1798 |
| Eisberg | 48088 | 9154 | 2100 | 1979 |
| Tuerkheim | 45576 | 10672 | 3044 | 2029 |
| Neuheilenbach | 47286 | 8731 | 3107 | 2197 |
| Dresden | 45787 | 9462 | 3122 | 2950 |

**Table 2.** Number of complete hourly non-rainy daytime radar echoes at the twelve selected radar stations.

|  | Jan | Feb | Mar | Apr | May | Jun | Jul | Aug | Sep | Oct | Nov | Dec |
|---|---|---|---|---|---|---|---|---|---|---|---|---|
| 2007 | 12 | 1 | 34 | 191 | 446 | 797 | 293 | 147 | 81 | 90 | 11 | 1 |
| 2008 | 3 | 1 | 24 | 13 | 212 | 808 | 1333 | 124 | 87 | 38 | 6 | 11 |
| 2009 | 16 | 12 | 38 | 26 | 264 | 209 | 1379 | 892 | 406 | 66 | 0 | 34 |
| 2010 | 36 | 73 | 52 | 45 | 74 | 541 | 1684 | 171 | 84 | 5 | 3 | 24 |
| 2011 | 2 | 31 | 13 | 145 | 210 | 716 | 741 | 190 | 139 | 59 | 2 | 7 |
| 2012 | 10 | 27 | 44 | 22 | 211 | 289 | 750 | 301 | 91 | 16 | 3 | 2 |
| 2013 | 53 | 18 | 53 | 46 | 65 | 318 | 1740 | 509 | 93 | 7 | 2 | 1 |

two urban (Essen, Frankfurt) and two coastal locations (Emden, Rostock). The twelve remaining radars give us roughly 20 thousand individual hourly images for comparison with COSMO-REA2. When studying the diurnal cycles below, we will furthermore include radar data at the full 5 min resolution which gives us over 200 thousand images.

In table 2, we see that the vast majority of clear sky echoes occurs during summer, particularly June and July, with considerable variability between the years. The preference for the warm season is expected since both insect activity and boundary layer






height are increased by higher temperatures. Consequently, the daytime frequency of available data follows a diurnal cycle as well (figure 3). In addition, there is a large second population of night time cases. The sudden increase in clear air echoes at dusk, as well as their absence in winter, hints at migrating swarms of insects as a likely explanation (Drake and Reynolds, 2012). We exclude these data because (1) the weaker nighttime convergences are less likely to influence the pattern of the
insect cloud and (2) migrating swarms tend to inhabit thin layers near an atmospheric inversion which only partly intersect the radar beam (cf. p.237 f. in Drake and Reynolds (2012)).

## 4 Methods

### 4.1 Wavelet analysis

The idea of this study is to compare the correlation structures of the radar reflectivities and divergence fields, summarized
in terms of scale, anisotropy and direction. To extract these properties from divergence and reflectivity images, we use the SAD methodology of Buschow and Friederichs (2021): The image to be analyzed is convolved with a series of localized 2D wave-forms with varying scale and orientation. The analyzing filters are the so-called *daughter wavelets* which are generated by shifting, scaling and rotating a single, carefully designed wave function, the *mother wavelet*. The square of one wavelet coefficient, i.e., the result convolving the image with one of the daughters, represents the amount of variance present at a
particular location for a particular combination of spatial scale and orientation. The dual-tree complex wavelet transform (Selesnick et al., 2005) used in this study provides daughter wavelets with six orientations and up to $J$ scales for an image of size $2^J \times 2^J$. Following Buschow and Friederichs (2021), the largest three scales are removed because their support is larger than the image, rendering their interpretation ambiguous. After spatial averaging, a radar image with $128 \times 128$ pixels is thus summarized by $4 \times 6$ values, the so-called wavelet spectrum. To extract the scale, anisotropy and direction from this spectrum,
we treat the $J \times 6$ values as point-masses arranged in a 3D space such that the six directions for one scale are at the vertices of a hexagon in the $x - y$-plane and the hexagons for the $J$ scales are located at $z = 1, \ldots, J$. The centre of mass of these point masses has three components in cylindrical coordinates:

- The central scale $z \in [1, J]$ measures the dominant spatial scale of the image. If all variance was at spatial scale $j$, then $z = j$; if all scales contain equal variance, then $z = (J - 1)/2$.

- The radius $\rho \in [0, 1]$ describes the anisotropy. If all directions have equal variance, then the centre of mass is in the middle of the hexagon and $\rho = 0$; if all energy is concentrated in one direction, then $\rho = 1$.

- From the angular coordinate, we can determine the dominant orientation angle $\varphi \in [0°, 180°]$. Note that $\varphi$ is only meaningful if the anisotropy $\rho$ is non-zero.

For a detailed description of the calculation of these properties, as well as the details of the wavelet transform itself, we refer
to Buschow and Friederichs (2021) and references therein. The software for this analysis is freely available in the open source `dualtrees` R-package (Buschow et al., 2020).





The central scale $z$ is a dimensionless quantity which cannot be analytically transformed into an equivalent Fourier wavelength. Since the actual physical size of the patterns is of some interest in the present study, we derive an empirical relationship based on test images with fixed wavelength in appendix A. We find that, in the range of $1.5 < z < 2.5$, the relationship is well
described by a linear fit with

$$\lambda \approx z \cdot 9\,km - 5.4\,km \tag{2}$$

It is important to note that this relationship is only approximately valid for the specific wavelets, scales and wave-like test images used in the present study. This equivalent wavelength is furthermore not identical to the spacing between wave-crests used as the measure of horizontal scale by Banghoff et al. (2020) because our $\lambda$ includes also the scale perpendicular to the
orientation of the features.

To make the distribution of angles $\varphi$ interpretable, we compute the angles of intersection between $\varphi$ and the model wind direction (averaged over the regions around each radar). A relative angle $\Delta\varphi = 0°$ thus means that the patterns align with the wind direction whereas $\Delta\varphi = 90°$ indicates an orthogonal orientation.

### 4.2    Boundary conditions and pre-processing

The wavelet analysis described above requires data on a regular grid, ideally of size $2^n \times 2^n$ to ensure fast computation times, discontinuities at the boundaries must be avoided. This is only a minor factor for intermittent fields like rain but very important for data with non-zero values along each border. To achieve periodic boundaries, we cut out a $128\,km \times 128\,km$ region (128 and 64 pixels for RADOLAN and COSMO-REA2, respectively) around each radar location and apply a circular Tukey window to smoothly reduce the field to zero (for divergences) or $-10\,dBz$ (for reflectivities) towards each side. A rectangular boundary
would introduce spurious horizontal and vertical directions to the wavelet spectra.

For the reflectivity data, further pre-processing steps are required. Firstly, some radar images contain erroneous isolated pixels with unusual intensities which would artificially reduce the analyzed spatial scales. Following Lagrange et al. (2018), we therefore compare each pixel to the average over its eight nearest neighbours. If the difference is greater than $10\,dBZ$, the pixel value is replaced by the neighbourhood average. Secondly, the reflectivities around each radar often contain gaps of very
small reflectivities ($< -10\,dBZ$), caused for example by buildings, mountains or water bodies without scattering insects. These arbitrarily shaped holes introduce an artificial pattern which is unrelated to the wind field and needs to be removed. Here, we fill in the gaps with a simple algorithm which iteratively replaces values below -$10\,dBZ$ with an average over the neighbouring non-missing pixels. The details of the gap-filling algorithm, as well as a demonstration of its effectiveness are given in appendix B.

Lastly, a comparison between the wavelet spectra of two images would normally require that both data sets be given on the same grid. In our case, we can avoid re-gridding either field since the spatial resolutions differ by a factor of two. The second scale in RADOLAN thus corresponds to the first scale of COSMO-REA2. We can therefore simply remove the smallest scale from the radar image to make the spectra comparable. We have checked that the results are virtually identical when the radar





**Figure 4.** Average central scale (a), anisotropy (b) and angle relative to the mean wind (c), calculated from COSMO-REA2 (2007-2013) in the environment of the selected radar stations. White contours mark the sun's elevation angle at $0°, 20°, 40°, 60°$.

images are bilinearly re-mapped instead. The largest daughter wavelet that fits into our domain is $j = 4$ for RADOLAN and

$j = 3$ for COSMO-REA2, giving us three comparable scales with six directions each.

## 5   Results

### 5.1   Climatology of divergence structures in COSMO-REA2

Based on section 2, we can expect that small-scaled, cellular circulations will form on warm sunny days, favored by high pressure and low wind speeds. Following the diurnal cycle of the boundary layer depth, these circulations start out small and

become larger over the course of the day. According to Poll et al. (2017), Banghoff et al. (2020) and references therein, we furthermore expect to see a more linear mode of organization on windier days. The orientation of these roll-like structures will generally follow the mean wind direction (Weckwerth et al., 1997). Both cells and rolls should leave an imprint on the scale and anisotropy and direction of the horizontal divergence fields. We therefore cut out square regions of $64 \times 64$ pixels around the twelve selected radar stations (table 1) and apply the wavelet analysis described above for all hourly time-steps from 2007

to 2013.

     As a first overview, we average the scale $z$, anisotropy $\rho$ and direction relative to the mean wind $\Delta\varphi$ over the hours of the day and weeks of the year. Figure 4 shows that all three simulated variables undergo pronounced diurnal and annual cycles. During nighttime, the average central scales of the divergence fields remain close to $z \approx 2$ (about $13\,km$) with no strong variations between seasons. After the solar elevation exceeds roughly $40°$, $z$ approaches a clear minimum around noon before increasing

again during the afternoon. This region of small values is surrounded by a ring of increased scales a few hours after sunrise and around sunset. These largest average scales coincide with a similar ring of unusually low anisotropy (figure 4 b). $\rho$ reaches a maximum during the early hours of the small-scale phase before decreasing during the afternoon. Concerning the orientation of the divergence field (panel c), we observe that the small-scale pattern is typically aligned with the mean wind direction while the larger scaled nighttime patterns are not.

As expected, the simulated small-scaled circulations thus impress their diurnal life-cycle on the mean spatial structure of the divergence field. To see how prominent these features are, compared to the overall variability, we now consider probability densities of the three structural quantities, separated by season and time of day (figure 5).

     For the spatial scales in panel (a), we find that the prominent minimum around noon is indeed a common occurrence in all seasons except winter, indicated by bi-modal distributions between 9 and 15 UTC. During summer in particular, the smaller-

scaled mode, centred near $z \approx 1.75$ or $\lambda \approx 10\,km$, is more likely than $z > 2$. Two modes can be seen with similar clarity in the distribution of orientations (figure 5 c): During winter or nighttime, orientations along the wind direction are rare, most angles are closer to $-75°$. In the other three seasons, $\Delta\varphi \approx 0$ is by far the most likely value during daytime. The signal in the



**Figure 5.** Estimated probability densities (kernel estimates) for the scale $z$ (a, converted into an approximate wavelength $\lambda$ via equation 2), anisotropy $\rho$ (b) and relative angle $\Delta\varphi$ (c) for different seasons and times of day.





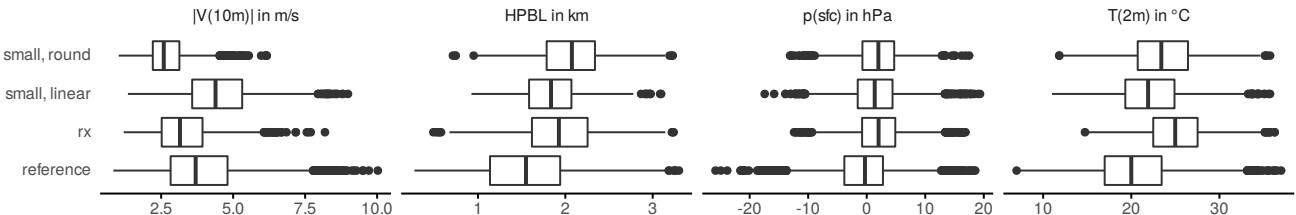

**Figure 6.** REA2 wind speed, boundary layer height, surface pressure anomaly and 2 m temperature during summer (JJA) between 11 UTC and 13 UTC, averaged around the selected radar locations. "small and round" cases have $z < 1.86, \rho < 0.12$, "small and linear" is $z < 1.86, \rho > 0.32$. The boxplot labeled "rx" contains all instances where at least one clear air radar echo is available.

anisotropy (figure 5 b), on the other hand, is far weaker: A clearly increased likelihood for anisotropic features is only evident in summer between 9 and 12 UTC and the change in the distribution is far less pronounced than for $z$. While the formation of exceptionally small structures, oriented along the mean wind, is thus a common occurrence, the increased linearity around noon seen in figure 4 b can only occasionally be observed.

Next, we are interested in the typical weather situation associated with the occurrence of these small and / or linear patterns. To this end, we focus on the three hours around noon during the summer season and search for cases where both $\rho$ and $z$ are in the bottom 5 % of their climatological distribution ("small and round" mode). For the "small and linear" mode, we select those cases where $z$ is in the bottom 5 % whereas $\rho$ is in the top 5 % of its distribution. At these time-steps, as well as the remaining "reference" cases, we compute spatial averages around the selected radar stations for several relevant variables from COSMO-REA2.

Figure 6 shows that boundary layer height, 2 m temperature and surface pressure see a moderate increase during time-steps with small and linear patterns and a stronger increase if the pattern is small and round. In the latter cases, the median temperature is close to $25°C$ and the boundary layer rarely falls below 2 km. Simultaneously, the average 10 m wind speed is strongly reduced. Conversely the small and linear mode is associated with a significantly increased wind speed. Hence the boundary layer circulation in COSMO-REA2 qualitatively resembles Rayleigh Bénard convection.

In preparation for the quantitative comparison with radar data, figure 6 also includes the environmental conditions for days where at least one clear-sky RADOLAN image is available. We find that the radar echoes occur mostly on very warm days with moderately increased boundary layer depth and decreased wind speeds. This is consistent with the assumption of insects as the primary origin of these echoes. The observations thus mostly sample cases where small-scale circulations are likely to occur.

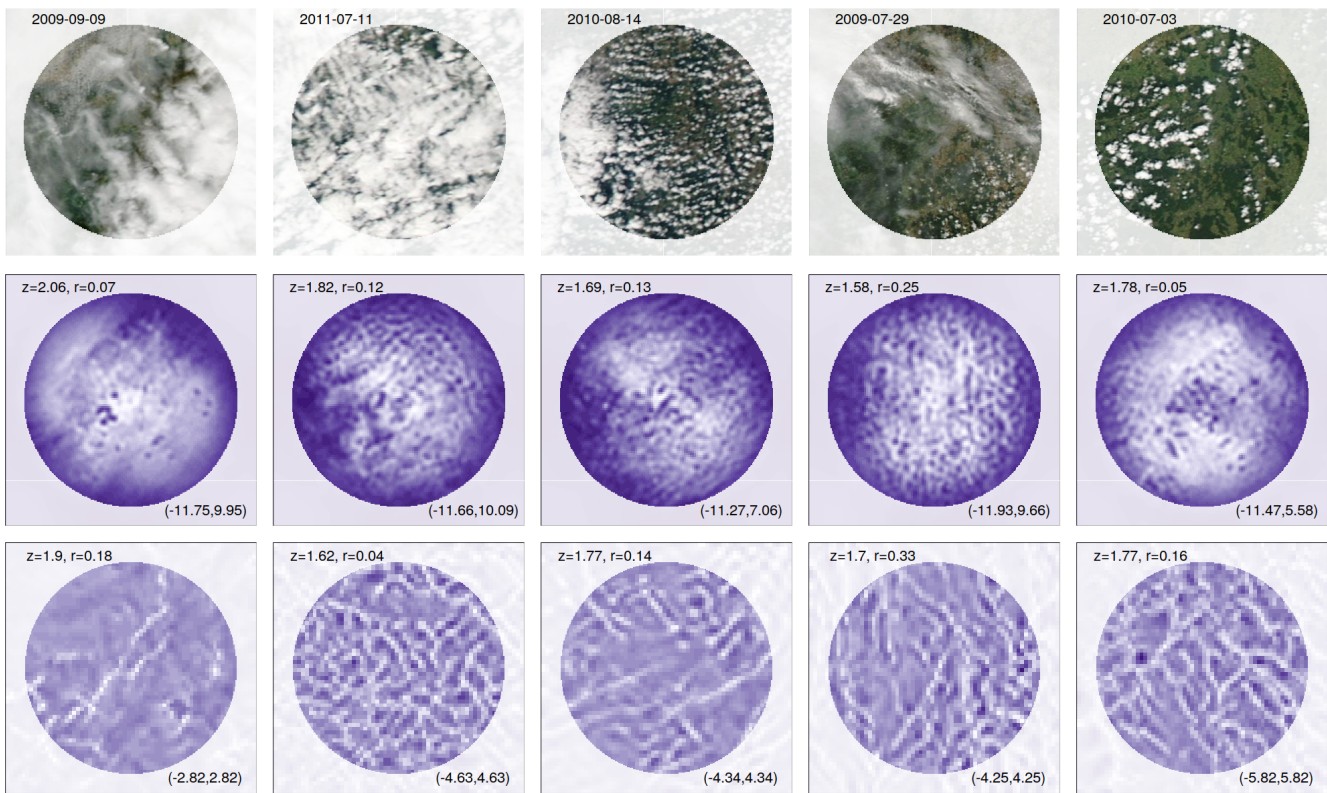

**Figure 7.** Randomly selected examples from the set of available, non-rainy 12 UTC radar images at Flechtdorf. Top row: Aqua MODIS snapshots (wvs.earthdata.nasa.gov, timing only approximately matches 12 UTC). Middle: RADOLAN RX reflectivity. Bottom: COSMO-REA2 10 m divergence. Light colors indicate high reflectivity and convergence, respectively. Numbers in the top left corner indicate the analyzed scale and anisotropy, the range of reflectivity / divergence values is given in the bottom right.

## 5.2 Verification against radar reflectivities

In this section, we attempt to assess the realism of our model-based climatology using the clear-sky radar reflectivity data from

RADOLAN. Besides cases with too many missing or rainy pixels, we also exclude all nighttime images. The remaining data is subjected to the wavelet analysis as described in section 4.2.

Before analysing the statistics of the entire dataset, we briefly consider a few individual examples. Figure 7 shows five randomly selected cases from the Flechtdorf radar station. The 12 UTC time step was chosen so that a visible satellite image from MODIS is available at approximately the same time. For consistency with the wavelet-based analysis, we have removed

the smallest-scaled features from the RADOLAN images by transforming to wavelet-space, setting the coefficients at level 1 to zero and transforming back.

The first two examples (leftmost columns) feature a closed cloud-cover; model and observation agree on a relatively large structure on 2009-09-09 and small, isotropic cells on 2011-07-11. The remaining three cases are all relatively small in scale



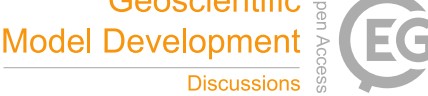

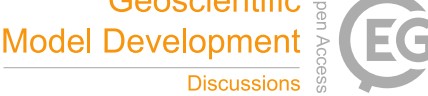

**Figure 8.** Diurnal cycles of spatial scales from 5 min radar data (areas and lines) and hourly COSMO-REA2 10 m divergence (points and error bars). Grey area and error bars indicate inter-quartile range, white line and black dots mark the median. Only cases with complete (see section 3.3) clear air echoes are included.

with both data-sets agreeing that 2009-07-29, i.e., our example from figure 2, has the smallest and most anisotropic structure.

Overall, the decent visual similarity between COSMO-REA2 and RADOLAN is reflected in small to moderate differences in $\rho$ and $z$.





Figure 8 shows a quantitative comparison of the modelled and observed diurnal cycles of central scales. In addition to the hourly data for which corresponding COSMO-REA2 divergences are available, we have included all other 5 min time-steps with complete clear-air echoes as well. The results can be separated into two main groups: At the rural radar stations in Eisberg,

Flechtdorf, Neuhaus, Neuheilenbach, Türkheim and Ummendorf, the agreement betweeen model and observations is surprisingly good. COSMO-REA2 reproduces not only the correct evolution of the diurnal cycle but also similar spatial scales with a large overlap in the inter-quartile ranges. In contrast, the observed spatial patterns at the three largest German cities of Berlin, Hamburg and Munich, differ significantly from the modelled values, as well as from the other stations. Hannover and Dresden have more data than the other urban locations (cf. table 1) and show better agreement with the model. Here, the observed

cycle is flatter but resembles its modelled counterpart in the afternoon. The unusual behaviour of the Feldberg/Schwarzwald station is likely the result of its mountainous surrounding which causes both additional ground clutter and changes to the local circulation, neither of which is resolved by the 2 km model orography. It is however worth noting that, despite the offset, both data sets agree that the smallest-scaled patterns occur later in the day than at other stations.

Good agreement between model and observations can be seen in the distribution of the angle $\varphi$ as well. In figure 9, we have

pooled all radars together and consider only full hours where the model wind direction is known. Cases with small observed anisotropy ($\rho \leq 0.1$), i.e., ambiguous orientation, were removed as well. We find that, between 10 and 17 UTC, both sets of images are usually oriented within $\pm 15°$ of the mean model wind direction; the distributions of RADOLAN and COSMO-REA2 match almost perfectly. Before and after this interval, which coincides with the small-scale phase of the diurnal cycle, a wider variety of orientations is possible.

While the scale and orientation are thus in reasonably good agreement, the same can not be said for the anisotropy. Figure 10 shows that the observations are almost universally more isotropic than the model fields. The pattern of increasing linearity towards a maximum before noon seen in figure 4 b is clearly present in this sample of the model data. The observations, on the other hand, hardly contain this pattern at all with only a very weak maximum at 11 UTC and nearly constant values during the afternoon.

Aside from the climatological distribution and diurnal cycle, we are interested in the model's ability to represent the day-to-day variability of the spatial divergence patterns. For $z$ and $\rho$, we can eliminate the overall bias and diurnal behaviour by subtracting the long-time mean for every daytime hour from the respective time series. To avoid residual effects of the annual cycle, we limit this analysis to the summer season. Timing errors within each day are furthermore removed by taking the daily minimum of $z$ and maximum of $\rho$. Figure 11 a reveals that the remaining scale anomalies in COSMO-REA2 and RADOLAN

are slightly correlated with many remaining errors below 0.1 and almost all below 0.2 (outer lines). As expected, the correlation is even lower for $\rho$ (figure 11 b) and the typical errors are relatively large even after the bias has been removed.

## 6 Discussion

The results of section 5.1 and 5.2 raise several intertwined questions: What level of realism can be expected of the reanalysed small-scale structure? To what extent can the RADOLAN data-set be used to validate the simulation? How appropriate was the



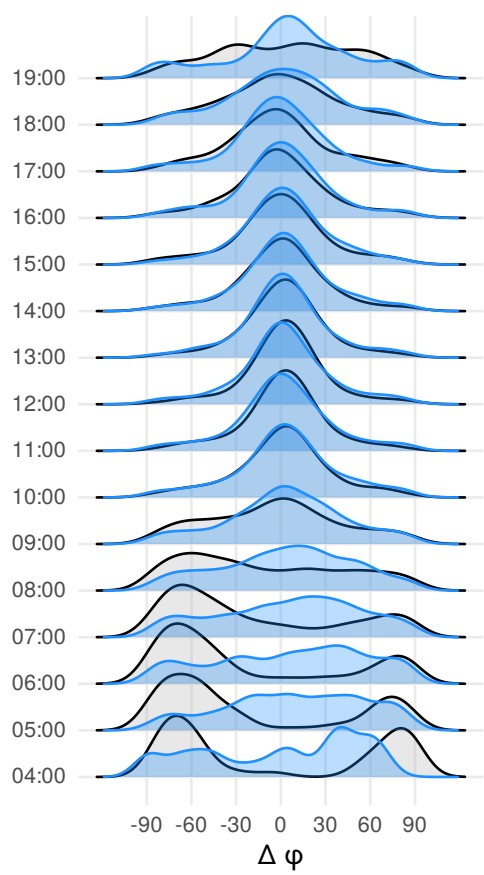

**Figure 9.** Distribution of orientations relative to the COSMO-REA2 mean wind throughout the day. COSMO-REA2 shown in black, RADOLAN in blue. Only complete, on-rainy daytime cases with $\rho(\text{RADOLAN}) > 0.1$ are included.

wavelet-based analysis for the task at hand? Concerning the trustworthiness of COSMO-REA2, it must be remembered that the local divergence patterns are primarily the product of the model dynamics and parametrized turbulence, not the data assimilation. The environmental conditions which drive the formation of a particular mode of small-scale organization, however, can be expected to have good accuracy due to the continuous input of wind speed, humidity and pressure from weather stations. It is therefore not surprising that the model can accurately represent diurnal and annual cycles and differentiate between days with

organized and unorganized situations. Consequently, the model climatology as described in section 5.1 qualitatively agrees with our expectations from the literature. Whether or not the simulated small-scale structure can itself be trusted is questionable in light of the theory discussed in section 2. Our comparison with RADOLAN clear-air data suggests that, despite the proximity to the Grey-Zone, the modelled structures are not overall unrealistic. In interpreting this result, we must recall that the difference in native resolution between RADOLAN and COSMO-REA2 was handled by deleting the smallest scale from

RADOLAN. We have thereby filtered out any variability below the model's effective resolution. Figure 8 therefore does *not*

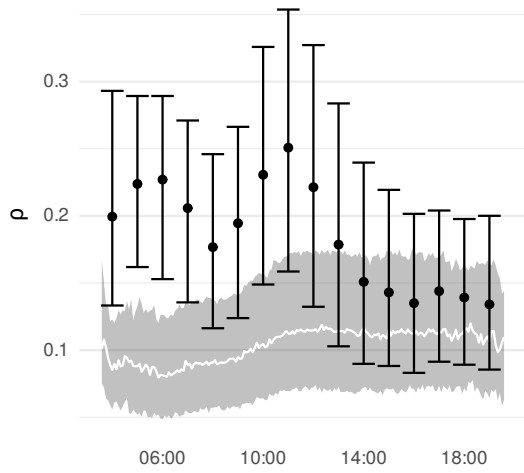

**Figure 10.** As figure 8, but for $\rho$ and without separation by radars.

indicate that the mesoscale model successfully simulates the spatial scales present in the real atmosphere. We can merely see that the *remaining* variability (upwards of $\lambda \approx 8\,km$), which both data-sets *can* represent, matches the observed diurnal cycle decently, especially at the rural stations.

As predicted by Zhou et al. (2014), the wavelengths of the simulated eddies are near the smallest scale resolved by the
model. We note, however, that the underlying resolution of RADOLAN is $1°$ in azimuth- and $250\,m$ in range-direction. Inside our $64\,km$ radius, and particularly close to the radar, the internal resolution of the measurements is considerably finer than the used $1\,km \times 1\,km$ grid. There is thus no obvious technical reason why, after filtering, RADOLAN should have to exhibit increased variability on the same scale as the model. We have experimentally re-calculated the central scales of the radar images including the previously removed smallest scales and found a slight shift in the cycle towards earlier hours. Conversely, if we
remove the second smallest scale as well, a shift in the opposite direction emerges. This supports our interpretation that the model simulates the patterns seen in the observations with an approximately correct diurnal cycle, *on the scales we included*; smaller-scaled variability, which would initiate earlier in the day, is resolved by neither COSMO nor RADOLAN. It should furthermore be noted that we make no direct statements about the intensity (variance) of the circulations. Such information cannot easily be inferred because the absolute radar reflectivities depend on the technical details of the radar, applied pre-
processing and the unknown overall concentration of biological scatterers.

The greater disagreement at the urban radar locations has two main explanations. On the one hand, it is likely that buildings and unrelated radio signals introduce excessive noise into the images, overshadowing the natural signal. This explanation is supported by the lack of complete images at the Essen and Frankfurt stations, both of which are located in highly urbanized regions (Frankfurt is the city with the most skyscrapers in Germany). On the other hand, the urban landscape itself can influence

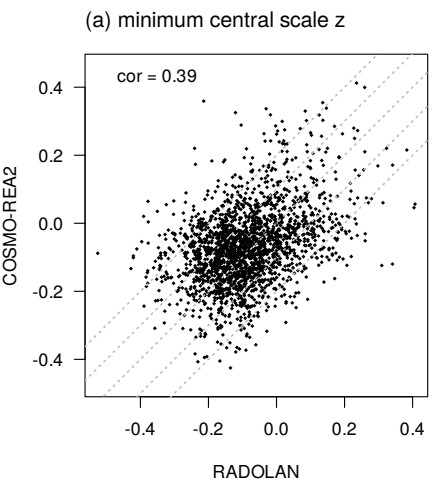

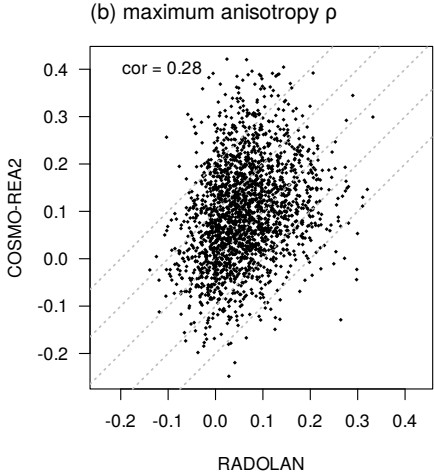

**Figure 11.** Scatter-plot of daytime minimum scale (a) and daytime maximum anisotropy (b) anomalies during daytime in summer (JJA) from RADOLAN (x-axis) and COSMO-REA2 (y-axis). Anomalies were calculated by subtracting the respective mean values from every hour of the day. Dashed lines mark errors $dz, d\rho = \{-0.2, -0.1, 0, 0.1, 0.2\}$.





the near-surface circulation in ways which are not resolved by the model. The similar effects of small-scale orography likely explain the special behaviour at the Feldberg/Schwarzwald station.

     Aside from spatial scales, the anisotropy of the divergence pattern, i.e., the difference between linear and cellular organization, is of interest. Here, the model's tendency towards more linear patterns earlier in the day could not be confirmed observationally. On the one hand, it is plausible that the lack of finer-scale variability leads to the simulation of unnaturally

regular stripes. On the other hand, gaps and noise have a larger impact on the anisotropy than the scale (cf. appendix B), making these results somewhat less robust.

     Lastly, it should again be emphasized that our clear air data-set provides no information on nighttime and winter and is biased towards cases with high temperatures where small-scale circulations are likely to occur. Our validation is therefore mostly conditional on the occurrence of these phenomena; whether or not the model correctly differentiates between days with

and without organized shallow convection could only partly be judged (cf. figure 11).

## 7    Conclusions and outlook

The main goal of this study was to explore the use of clear-sky radar data for the evaluation of simulated low-level divergence structures. A wavelet-based verification methodology, developed and extensively tested for precipitation data, was used to summarize the spatial patterns in terms of scale, anisotropy and direction. We have demonstrated that model-based divergences

and radar reflectivities are comparable at this level of abstraction. Our investigation of the German radar network has shown that usable clear sky echoes are rare overall and almost non-existent in winter. This supports the assumption that such daytime echoes are caused by small insects, the life cycle and habitats of which may also explain the substantial differences between radars as well as strong year to year variations. The relatively long time-span from 2007 to 2013 nonetheless resulted in a robust data set of over 20.000 individual images, mostly during summer, where the modelled patterns could be verified against spatial

observations. At most radar locations, both data sets show a very similar diurnal cycle in the spatial scales and orientations with a strong preference for small-scaled ($\lambda \approx 10\,km$) features around noon. The orientation during the small-scaled phase of the cycle is almost always within $15°$ of the mean wind. The fact that this observation holds for both data sets also implicitly confirms that the model adequately represents the mean wind direction. COSMO-REA2 furthermore simulated a trend towards increasingly linear features at the start of the small-scaled phase which could not be found in the observations. As discussed

above, a more complete set of observations might be able to clarify whether this indicates deficiencies of the model or the observations or (likely) both.

     Based on the overall decent agreement with the radar observations, we may put some trust in the model's behaviour at the unobserved parts of the time series as well. If COSMO-REA2 is thus to be believed, mesoscale shallow convection, favored by high pressure (clear skies) and temperatures, as well as weak winds, is a common occurrence in Germany in all seasons

except winter; during JJA, the small-scale mode is more likely than the larger-scaled configuration. Its onset a few hours after sunrise is characterized by a transition phase with larger scaled, isotropic divergence patterns, the orientation of which switches from $\sim 70°$ to $\sim 0°$ with respect to the mean wind direction. While most patterns are isotropic, i.e., cellular in nature, there





is also a weaker signal of linear organization. This more roll-like mode is most often simulated during JJA between 9 and 12 UTC and preferably occurs when winds are unusually strong and the boundary layer is shallower than in the cellular cases.

These simulated features are qualitatively consistent with the theory, as well as previous observations of mesoscale shallow convection.

Concerning future prospects, it must be emphasized that we have relied on only the most widely available kind of radar observations. Modern dual-polarization Doppler radars produce a wealth of further information, which would for example allow us to confidently separate insect-related echoes from unhelpful noise and clear up the nature of the night-time echoes

(Zrnic and Ryzhkov, 1998; Melnikov et al., 2015). Additionally, parameters like mean wind speed and direction, and even the boundary layer height (Banghoff et al., 01 Aug. 2018) could be inferred directly from the radar instead of relying on the model (Banghoff et al., 2020). Lastly, we re-iterate that small scales below $\sim 8\,km$ were filtered out in this study in order to fairly evaluate the mesoscale model. Depending on their frequency, weather radars can observe much finer details of the turbulent boundary layer. A similar strategy to ours could therefore also provide useful information for the objective validation

of realistic large eddy simulations as in Thurston et al. (2016); Poll et al. (2017); Bauer et al. (2020); Ito et al. (2020); Pantillon et al. (2020).

*Code and data availability.* Software for the dual-tree wavelet transformation is available in the `dualtrees` R-package (Buschow et al., 2020). In addition, the specific version (0.1.4) used for this manuscript has been permanently archived at https://doi.org/10.5281/zenodo. 5027277 (Buschow, 2021a). COSMO-REA2 is currently available from the website of the Hans Ertel centre (reanalysis.meteo.uni-bonn.de).

RADOLAN is available via the DWD OpenData portal (opendata.dwd.de). The cropped reflectivity and divergence fields around the used radar station have been archived at https://doi.org/10.5281/zenodo.5036447, together with all auxiliary data and software needed to fully reproduce the figures in this manuscript from scratch (Buschow, 2021b).

## Appendix A: Empirical relationship between scale and wavelength

To approximately translate the central scale into an equivalent Fourier wavelength $\lambda$, we apply the exact method described in

section 4.2 to synthetic test images of pure sine-waves, given by

$$f(x,y) = \sin\left(2\pi(k_x x + k_y y)\right) + \epsilon$$

where $\epsilon$ is a Gaussian white noise term with zero mean and variance 0.04. Figure A1 shows that the relationship between $z$ and $\lambda$ is nearly linear for this idealized signal. For $z < 1.5$ and $z > 2.5$, the curve becomes non-linear because most variance is outside the range of scales covered by our wavelet transform. The linear fit yields $\lambda/\Delta_x = 4.464 \cdot z - 2.765$. Since we are

merely interested in a rough approximation with round numbers, we simplify the result for $\Delta_x \approx 2\,km$ to obtain equation 2.

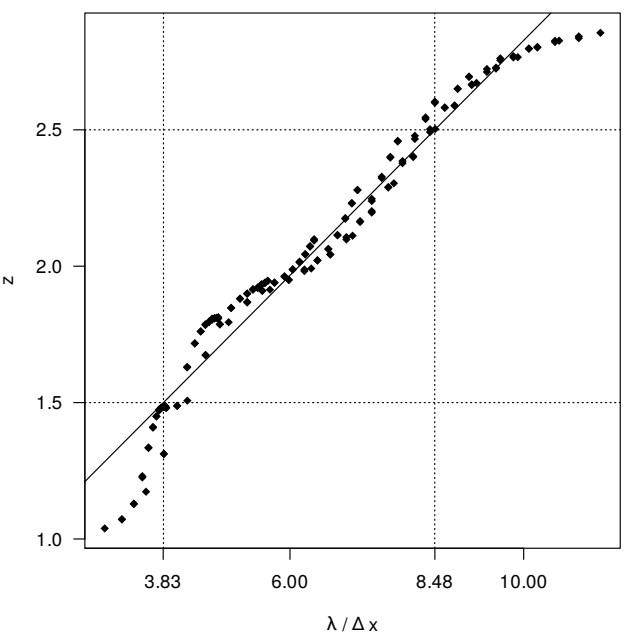

**Figure A1.** Wavelength $\lambda = 1/\sqrt{k_x^2 + k_y^2}$ against central scale $z$.

## Appendix B: Filling the gaps in the radar images

For this study, we are not interested in the radar reflectivities themselves, or even their full spatial correlation function, but only the estimates structural characteristics $\rho, \varphi, z$. To mitigate the effects of holes, i.e., regions with $Z \leq -10\,dBz$, in the radar images, we implement a simple iterative algorithm to smoothly fill in the gaps: (1) Find missing points with at least one non-missing neighbour, (2) replace values of those points with an average over the up to eight adjacent non-missing values and (3) repeat from (1) until all gaps are filled. The result is similar to inverse distance interpolation but (at least in our implementations of the two algorithms) considerably faster. To test the success of our approach, we select 300 nearly complete (less than 3 % missing data) clear-air radar echoes from our data-set and artificially add the gaps form 300 other randomly selected incomplete images. In figure B1, we compare $\rho, \varphi, z$, estimated with and without the gap-filling algorithm. As expected, the impact of the gaps is massive but our algorithm mostly mitigates the effects. We have repeated the experiment with inverse distance interpolation (not shown) and found no substantial improvement over the iterative procedure.

*Author contributions.* SB had the idea for this work, both authors jointly developed the original methodology. Writing and coding was led by SB, with suggestions and additions from PF. Both authors contributed to the final draft and proof-reading.



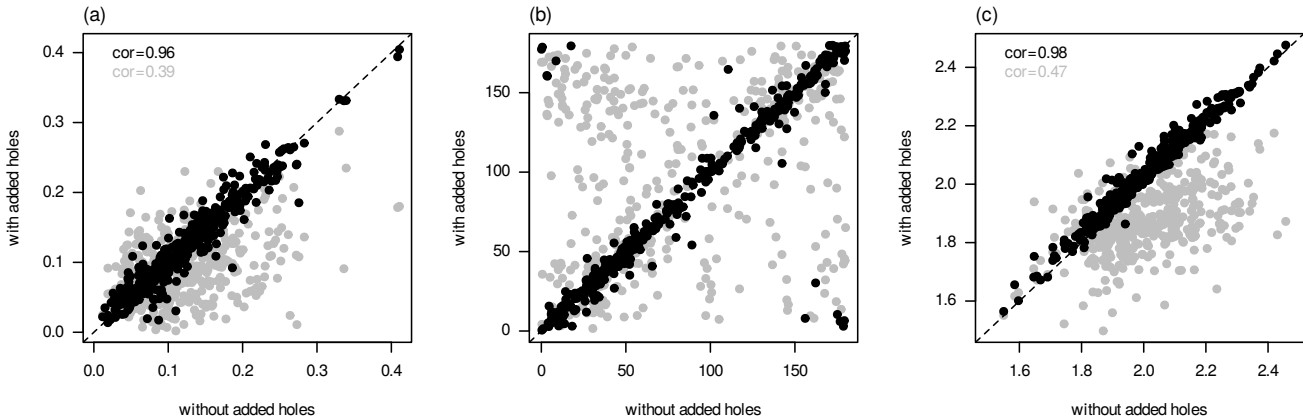

**Figure B1.** Anisotropy $\rho$ (a), angle $\varphi$ (b) and scale $z$ (c) estimated from nearly complete images (x-axis) and images with added holes (y-axis). Black dots show the results of the iterative gap-filling algorithms; values obtained without gap filling are shown in grey.

*Competing interests.* The authors declare that they have no conflict of interest.



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
