# Peer review of "Verification of Boundary Layer Wind Patterns in COSMO-REA2 using Clear Air Radar Echoes"

_Geoscientific Model Development, 2021_

## Author Response (AR1)

Thanks for the suggestions and constructive criticism! All of our individual answers are listed below, the corresponding changes have been color-coded in the manuscript attached at the end.

**Chief editor comment**

*As your analysis is on COSMO-REA2 data only, please add something like "a case study using COSMO-REA2" to the title of your manuscript.*
We have replaced "Germany" by "COSMO-REA2" in the title.

**Reviewer comment #1 (marked up in magenta)**

*However, as described in specific comments, the motivation to examine the divergence of horizontal winds is unclear. This manuscript may be acceptable after a considerable revision. The atmosphere at the height of 10 m is generally in the surface layer. It sounds strange that divergence in the horizontal winds in the surface layer are used to characterize the larger scale structures that span the entire boundary layer.*
*Authors should reconsider a variable that physically relates to the organized structures in the boundary layer, while variables archived in the reanalysis dataset may be limited.*
*If authors wish to use the present variable, it may be better to examine surface observations regarding the variations in the divergence in the real atmosphere. Otherwise, investigating the divergence would illuminate an unrealistic aspect of the surface layer in the simulation.*
This is indeed a good point. In the real atmosphere, we would expect the 10m wind field to look very different from the overall boundary layer circulation. In this sense, the good agreement of the 2km model with the structures from the radar is actually unrealistic. We therefore follow your suggestion and replace the 10m divergence by the corresponding fields from model level 45 (height of approximately 200m) which is the highest readily available level of COSMO-REA2. This data lies above the surface layer and should make for a more appropriate comparison. We replaced "near surface" with "boundary layer" in the title as well. The decision to switch levels lead to a number of minor changes throughout the manuscript (replacing all mentions of 10m wind etc.) which were accordingly highlighted. For figure 5, we moved to the 10% quantiles because the joint tails of rho and z were now too sparsely populated at 5%. The results here are qualitatively unchanged.

*As mentioned above, the structures extracted from the reanalysis are more or less artificial. According to Table 2, validations by the clear Radar echos are possible only for the summer. The results shown in Section 5.1 should limit those that can be validated in Section 5.2.*
We trust that the paper makes it overall clear which part of the data-set could and could not be validated with radar data. We believe that there is then no harm in showing an overview of the simulated structure of the complete data-set, especially since we are not aware of any similar model-based long-term climatology of this particular kind. Our analysis of the observable part of the time-series shows that the structures are not, in fact, entirely "artificial", so the rest of the climatology may be of some interest as well.

*The image of fig. 4 does not appear in the manuscript (only its caption appears). I could not evaluate the discussion associated with the figure.* Figure 4 is now correctly included in the manuscript.

*In the visual appearances as Figs. 2 and 7, the horizontal scales of organized structures are different between the reanalysis and radar images. Figure 8 shows that the scale in the radar images is even larger. It would be better to add a clear explanation for this very counter institutive*

*result.* As already mentioned in the discussion (section 6), both COSMO and RADOLAN are of course missing some of the smallest-scaled variability that occurs in nature. In the interest of fair comparison, we have filtered out the smallest scale of RADOLAN (see section 4.2) and thus limit our validation to those scales that both could theoretically have in common. We have added another sentence to the discussion which explicitly relates this to the unintuitive observation in figure 8.

*Line 133 Is the lack of insects at the top of the boundary layer only due to the temperature?*
Migrating swarms of insects can sometimes be observed above the boundary layer as well. These species are typically larger and can, to some extent, resist the atmospheric flow to progress longer distances. Such migrations typically occur around dusk and dawn and were not further investigated here.

*Many acronyms are used without definitions. Please check throughout.*
Good point, there was indeed quite a number of undefined acronyms. Some were removed, all others should now have a definition.
- COSMO-REA2: The abstract already explains that this is a regional reanalysis, the acronym COSMO is now defined in the introduction.
- RADOLAN: Removed from the abstract, explanation moved to the first occurrence in the introduction.
- VERA, AROME, MesoVICT: None of these were actually needed in the context of this paper, we have removed them to reduce the overall number of acronyms.
- MODE, DAS, SAL, SAD, FSS: Replaced the acronyms by their full names as they only occurred once.
- NWP: Only occured once and was thus replaced by the full term.
- WRF, TerrSysMP: Added the definitions.
- LES, TKE: Replaced by the full term.
- DWD: Added the explanation "German weather service"

*Line 90: It should be described as λ~2d* Replaced "=" by the approximate sign.

*Figure 2 Units and explanations of shading in panel a are missing.* They have been added.

*Line 159: "+10dB" may be a mistake for "-10dB".* It is not, we identify rainy episodes by some number of pixels with relatively large values (compared to the clear air cases).

*Figure 6: Box plots and definitions of "rx" and "reference" are not clearly explained.* An explanation for "reference" was added to the figure caption. The label "rx" is now also mentioned explicitly at the appropriate point of the text.

*Line 268: I'm not sure the use of "see" is grammatically correct.* Replaced "see" by "undergo".

**Reviewer comment #2 (marked up in orange)**

*I have only one major comment about this study. Given that multi-Doppler wind retrieval techniques exist (Bousquet et al. 2008, and Beck et al. 2014 cited in this manuscript by the authors), why the authors did not retrieve two-dimensional or three- dimensional wind fields first, then compare the retrieved winds, or divergence product from multiple radars and compare with the regional analysis COSMO-REA2? Please explain why.*
Retrievals of horizontal wind fields require at least two overlapping radar beams. Within an

operational radar composite, this is generally only achieved at heights well above 1km, i.e., outside the boundary layer we are interested in (see for example figure 2 in Bousquet and Tabary (2014), doi.org/10.1002/qj.2163). Beck et al. 2014, for example, only look at 2km height to achieve decent coverage. They also limit themselves to rainy episodes, we are not sure how reliable multi Doppler retrievals from clear-air echoes might be. In addition, we are not aware of any such composite for Germany. The generation of such a data-set is far from trivial and would go way beyond the scope of this paper. This is why we use the more widely available reflectivity data instead. An explanation to this effect has been added to the introduction.

*Abstract: It's not good to use acronyms COSMO-REA2 and RADOLAN without explanation first. You may put these acronyms into a list and put it as an appendix?* We chose to remove some of the superfluous acronyms and explained the rest at the appropriate points in the text (see answer to reviewer #1).

*Page 3, line 66-67: Why not use the same large data-base of radial velocities to retrieve winds, then derive divergence structures and do comparison with the model reanalysis COSMO-REA2?* See above.

*Page 4, line 106-107: What are TerrSysMP and COMSO? Please explain.* See above, all acronyms are either removed or explained.

*Page 5, line 118: Please explain what is the "modified Tiedtke mass-flux scheme? You may need to add a reference here?* A reference has been added.

*Page 6, Figure 2: Please explain units used in the color bars? What do these numbers for the color bars represent?* Explanations have been added (see above).

*Page 11, Figure 4: is missed out from the manuscript.* Figure 4 is now included.

*Page 12, Figure 5: If DJF, MAM, JJA, SON represent seasons, please use English.* We have added a further explanation to the figure caption, but feel that the acronyms are rather common and widely understood.

*Page 15, figure 8: Please add units for both x- and y- ordinates.* The axes are now correctly labeled.

*Page 15, line 288-289: Rewording "The remaining three cases are all relatively small in scale with both data-sets agreeing that 2009-07-29…".* This section has changed in response to reviewer 3 (see below).

*Page 18, Figure 10: Please add "hours" for the x- ordinate.* Added it to Fig.9 as well for consistency.

*Page 20, line 374: Should "20.000 individual…" be "20,000 individual…"?* It seems that the preferred way of writing large numbers is "20 thousand".

*Page 23, Figure B1: What are the units for these x- or y- ordinates? Please explain the correlation numbers in the up-left corner of each image in the caption. Why (b) misses those correlation numbers?* Degree signs have been added to the angles in panel (b). Section 4.1 now explicitly mentions that the central coordinates are dimensionless and have no unit. An explanation for the correlations has been added to the caption. Linear correlation is not helpful for a circular quantity like the angles and panel (b). We feel that the scatter-plot nonetheless demonstrates the very convincing agreement.

**Reviewer comment #3 (marked up in cyan)**

*Fig. 7, Honestly, I found it difficult to compare the radar reflectivity patterns with the COSMO-REA2 10 m divergence. While the model data show coherent small-scale structures that are typical of the surface layer under convective conditions, the radar reflectivity field seems to be filled with much smaller "cellular" structures, and is somewhat noisy. Perhaps I am not well-trained in reading clear-sky radar echo, but I hope the authors could expand the discussion of Fig. 7, pick a case and help the readers compare these images qualitatively?*
The radar images are likely noisier by nature due to irregularities in the scatterer concentration, the local topography and other sources of measurement noise. We admit that the visual similarity in these randomly selected cases is not as great as our short description made it sound. We have changed the text accordingly.

*Line 140, did you mean "SW/NW" instead?* In fact, we meant SE/NW.

*Line 265, not sure what you mean by "at these time-steps".* We have made it clear that we refer to the time-steps that meet the "small and linear" or "small and round" criteria defined in the preceding sentences.

*Line 302-303, "It is however worth noting that, despite the offset, both data sets agree that the smallest-scaled patterns occur later in the day than at other stations", so what does such delay tell us?* We believe that this may simply be the effect of overall lower temperatures at the greater altitude.

*Line 374, did you mean you mean "20,000" rather than "20.000"?* Changed it to "20 thousand" (see above).

*Line 396, fix the reference "(Banghoff et al., 01 Aug. 2018)".* Fixed it. Also fixed a few other strange-looking entries in the bibliography.

---

## Author Response (AR2)

*Thanks again for the suggestions and encouragement!*

Please make sure that the units are not in italics. Use "\unit{}".
*Done.*

The abstract refers to the "German radar composite". I am not sure if everyone knows what this is.
*It seems that this term is indeed ambiguous (for example this is **not** what we mean:*
[http://okfirst.mesonet.org/train/nids/CREF.html](http://okfirst.mesonet.org/train/nids/CREF.html)*). It has been replaced by "mosaic", which should give everyone an intuitive idea.*

l23: single -> few?
*You probably meant line 32, we replaced "single" by "a few".*

Fig1, caption: Do not capitalize "Eddies"
*Done.*

Could you add a reference for the Tukey window?
*Done.*

In English language there is a difference between "which" and "that" many are not aware of. https://www.diffen.com/difference/That_vs_Which. E.g. "which" in lines 223 and 228 should be "that". This list is probably not complete.
*Instances of "which" and "that" throughout the manuscript have been checked and corrected to the best of my understanding.*